# Estimating the Burden of Illness of Relapsed Follicular Lymphoma and Marginal Zone Lymphoma in Ontario, Canada

John Kuruvilla [1,2], Emmanuel M. Ewara [3,*], Julia Elia-Pacitti [4], Ryan Ng [5], Maria Eberg [5], Atif Kukaswadia [6] and Arushi Sharma [7]

1. Department of Medical Oncology and Hematology, Princess Margaret Cancer Centre, 610 University Avenue, Toronto, ON M5G 2C1, Canada; john.kuruvilla@uhn.ca
2. Department of Medicine, University of Toronto, Toronto, ON M5S 3H7, Canada
3. Market Access, Janssen Canada Inc., 19 Green Belt Drive, North York, ON M3C 1L9, Canada
4. Medical Affairs, Janssen Canada Inc., 19 Green Belt Drive, North York, ON M3C 1L9, Canada
5. Real World Solutions, IQVIA, 16720 Rte Transcanadienne, Kirkland, QC H9H 5M3, Canada
6. Real World Solutions, IQVIA, 300-6700 Century Avenue, Mississauga, ON L5N 6A4, Canada
7. Real World Solutions, IQVIA, 535 Legget Drive, Tower C, 7th Floor, Ottawa, ON K2K 3B8, Canada
* Correspondence: eewara@its.jnj.com

**Abstract:** Background: Many patients with advanced follicular lymphoma (FL) and marginal zone lymphoma (MZL) relapse after first-line chemotherapy. Objective: To examine healthcare resource utilization (HCRU) and cost, treatment patterns, progression, and survival of patients with FL and MZL who relapse after first-line treatment, in Ontario, Canada. Methods: A retrospective, administrative data study identified patients with relapsed FL and MZL (1 January 2005–31 December 2018). Patients were followed for up to three years post relapse to assess HCRU, healthcare costs, time to next treatment (TTNT), and overall survival (OS), stratified by first- and second-line treatment. Results: The study identified 285 FL and 68 MZL cases who relapsed after first-line treatment. Average duration of first-line treatment was 12.4 and 13.4 months for FL and MZL patients, respectively. Drug (35.9%) and cancer clinic costs (28.1%) were major contributors to higher costs in year 1. Three-year OS was 83.9% after FL and 74.2% after MZL relapse. No statistically significant differences were observed in TTNT and OS between patients with FL who received R-CHOP/R-CVP/BR in the first line only versus both the first- and second- line. A total of 31% of FL and 34% of MZL patients progressed to third-line treatment within three years of initial relapse. Conclusion: Relapsing and remitting nature of FL and MZL in a subset of patients results in substantial burden to patients and the healthcare system.

**Keywords:** burden of illness; follicular lymphoma; marginal zone lymphoma; costs; epidemiology

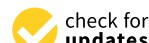



## 1. Introduction

The burden of cancer in Canada is significant on the health of both men and women and the Canadian healthcare system [1]. Non-Hodgkin's lymphomas (NHL) are a heterogeneous group of lymphoid neoplasms which include follicular lymphoma (FL) and marginal zone lymphoma (MZL), two indolent, slow-growing subtypes of NHL. FL accounts for 20%–30% of all NHL cases [2], with an estimated age-standardized incidence in Canada of 38.3 cases per million individuals per year. MZL comprises 7% of all NHL, with an estimated age-standardized incidence rate of 19.6 per million person years [3]. Survival with FL and MZL improved in recent decades. The age-standardized, five-year relative survival for FL increased from 58.9% to 74.3% between 1984–1993 and 2004–2013 [4]. Improvements in survival are attributed in part to advances in treatment including improvements in radiation therapy, new chemotherapy options, the introduction of the monoclonal antibody rituximab, and autologous stem cell transplantation [4–7].

In general, treatment is intended to relieve symptoms and delay relapse, and treatment choice depends on the aggressiveness of the tumor, age, and general health of the patient. The standard first-line treatment for symptomatic FL and MZL in Canada is chemotherapy in combination with the anti-CD20 antibody rituximab [8]. The most frequently used regimen for symptomatic, advanced-stage, first-line therapy is six cycles of rituximab-bendamustine or six–eight cycles of rituximab (R)-CVP (cyclophosphamide, vincristine, and prednisone), followed by eight cycles of rituximab maintenance therapy given every 3 months. In recent years, bendamustine plus rituximab (BR) became a preferred regimen in the first line setting because of better long-term disease control and favorable toxicity profile compared to R-CHOP [9,10]. The recommended treatment for older patients without organ dysfunction is rituximab monotherapy to avoid the toxicity associated with chemotherapy [11]. As per guidelines, suggested treatment regimens for patients who relapse after first-line treatment include re-treatment with a rituximab-based chemoimmunotherapy, lenalidomide in combination with rituximab, or other targeted therapies including ibrutinib (relapsed/refractory MZL only), ibritumomab tiuxetan, and Obinutuzumab and rituximab monotherapy for elderly or infirm patients [12,13]. However, access to these therapies depends on funding which can vary by jurisdiction and time period.

Currently, the literature on the impact and overall cost of relapsed FL and MZL is limited. A previous review of four studies published between January 2006 and November 2016, three from the USA and one from Denmark, estimated the cumulative total direct healthcare costs for FL to be higher for patients with progressive disease ($30,890) relative to those without ($8740) and identified the use of chemotherapy as the main driver of costs [14]. A retrospective, real-world evidence study of patients diagnosed with FL between 1 January 2010 and 31 December 2013 used MarketScan® data from the United States to estimate all-cause healthcare costs per patient per year that ranged from $97,141 (SD: $144,730) for first-line therapy to $424,758 (SD: $715,028) for fifth-line therapy [15]. Canadian-specific burden of illness literature for FL and MZL is scarce, making an informed evaluation of novel treatment strategies, with associated resource allocation decisions, challenging.

The overall aim of this study was to estimate healthcare resource utilization (HCRU) and associated direct healthcare costs for Canadian patients with FL or MZL who relapse after initial treatment, using administrative health data from Ontario, Canada. The secondary objectives were to describe patient characteristics, treatment outcomes, and treatment patterns in FL and MZL patients who progress to second- and third-line therapies.

## 2. Materials and Methods

### 2.1. Study Design

A retrospective, longitudinal, population-based study was conducted using administrative health data from Ontario, Canada. The study was designed to ensure adequate follow-up for the relapse population. Two separate cohorts were defined, one with patients who relapsed following first-line treatment for FL and another with patients who relapsed after first-line treatment for MZL, and who were newly diagnosed with either FL or MZL between 1 January 2005 and 31 December 2012, and relapsed before 31 December 2018. Patients were followed for up to three years post relapse (index event) until death, censoring due to histological transformation to diffuse large B-cell lymphoma (DLBCL), end of the 3-year study period, end of Ontario health insurance plan (OHIP) coverage, or 31 March 2020, whichever came first.

### 2.2. Data Sources

In Ontario, all medically necessary services are paid for by a single payer insurance system (OHIP). Healthcare utilization information was available from 1986 onward in the form of administrative records that are deidentified and linked using unique encoded identifiers housed at ICES (formerly the Institute for Clinical and Evaluative Sciences). Data were analyzed at ICES, a non-profit research institute whose legal status under Ontario's health information privacy law allows it to collect and analyze healthcare data without

consent, for health system evaluation and improvement under Sections 4 and 5 of Ontario's Personal Health Information Protection Act [16]. The Ontario Cancer Registry (OCR) was used to identify patients diagnosed with FL and MZL and their cancer diagnosis date. Demographic data, including sex, age, and residence were extracted from the registered persons database (RPDB). Neighborhood-level income quintile was derived using census data, based on the median income in each dissemination area [1], and linked to patients based on residential postal codes [17]. Treatment information was obtained from the new drug funding program (NDFP) and cancer activity level reporting (ALR) datasets. The discharge abstract database/same day surgery database, national ambulatory care reporting system and OHIP data were used to estimate HCRU and associated direct healthcare costs. Ontario drug benefit (ODB) data captured all prescription claims dispensed under Ontario's provincial public drug program.

### 2.3. Study Population and Selection Criteria

The study identified all FL or MZL cases who received R-chemotherapy as primary treatment, were newly diagnosed between 2005 and 2012, and relapsed prior to the end of 2018. FL and MZL patients had to have received first- and second-line treatment (the relapse event) from 1 January 2005 to 31 December 2018 to be included in the study cohort. FL and MZL patients were excluded if they met any of the following criteria: invalid OHIP card number, invalid or incomplete records (i.e., missing age, sex, missing other demographic information, age $\geq$ 105 years at diagnosis, death on the date of diagnosis), primary residency outside of Ontario, or had an undetermined FL or MZL diagnosis with different cancer diagnoses on the same date. Patients were also excluded if they did not receive any systemic treatment during the study period, received only first-line treatment during the study period (i.e., no relapse event), or systemic treatment was received through participation in a clinical trial; initiated a regimen outside of the recommended treatments for first- or second-line of therapy for FL or MZL, had refractory disease or treatment intolerance, or experienced transformation to DLBCL prior to the relapse event.

### 2.4. Study Outcomes

HCRU was assessed for multiple healthcare touchpoints, including general practitioner (GP) visits, oncologist and hematologist visits, all other specialist visits, hospitalizations, and emergency department (ED) visits. Direct health care costs captured the following publicly funded healthcare services: physician billings, inpatient hospitalizations, same day surgeries, ED visits, national ambulatory care reporting system (NACRS) visits to cancer clinics, ODB claims, NDFP chemotherapy drug costs and aggregated costs for other services. Estimated amounts were standardized to the 2019 Canadian dollar.

Duration of therapy was estimated as the time from the start of the chemotherapy regimen to the completion of R-maintenance. Progression-free time was estimated as the time from the last administration date of first-line therapy to the relapse event date. Overall survival and time to next treatment (TTNT) were estimated as the cumulative death incidence with transformation to DLBCL as a competing risk and cumulative incidence of starting third-line therapy with death and transformation to DLBCL as competing risks, respectively. Patients were followed from the last administration date of the second-line therapy until the first administration date of the third-line therapy or death, censoring due to histological transformation to DLBCL, the end of the analysis period, end of OHIP coverage, or 31 March 2020, whichever occurred first. Eligible third-line therapies were R-monotherapy, R-CHOP, R-CVP, BR, bendamustine with no rituximab, chlorambucil, etoposide, fludarabine, gemcitabine, or dexamethasone-high dose cytarabine-cisplatin (DHAP), cyclophosphamide, lenalidomide, ibrutinib, or idelalisib.

### 2.5. Stratification

Outcomes were stratified by treatment received in the first and second line. Patients were either categorized into having received rituximab-based therapy in the first line only

or both in the first and second line. Those who received rituximab-based therapies in the first line only received any of R-monotherapy, R-CHOP, R-CVP, or BR in the first line and regimens that included bendamustine with no rituximab, chlorambucil, etoposide, fludarabine, gemcitabine, or DHAP in the second line. Those who were classified as receiving rituximab-based therapy in both the first and second line received any of R-monotherapy, R-CHOP, R-CVP, or BR in both the first and second line.

### 2.6. Statistical Analysis

Patient characteristics were summarized as frequency and percent, *N* (%). Annual estimates of HCRU counts were presented as means by healthcare touchpoint. Direct healthcare costs were estimated using the person-centered costing methodology developed at ICES [18]. The total direct healthcare costs were calculated for each year over a patient's follow-up. Calculation of HCRU or costs ended for a relapse patient if he/she had a histological transformation to DLBCL, died, was lost to follow-up, or the study period ended. The proportion of patients with any HCRU and non-zero costs refer to having any one of a GP visit, an oncologist, or hematologist visit, any other specialist visit, an inpatient visit, or an ED visit. Data were extracted in December 2020 and analyzed using SAS v9.4 (SAS Institute Inc., Cary, NC, USA). Given the privacy rules regarding access to the data, all analyses were conducted by ICES staff. Small cell values were reported as a range according to ICES reporting standards to reduce the risk of re-identification. OS and TTNT were estimated with cumulative incidence functions (CIFs) to account for competing events (i.e., death, transformation to DLBCL) with Gray's test used to assess the statistical significance of the differences between CIFs [19].

### 2.7. Ethics

This study was designed and implemented with ethics approval from the Institutional Review Board Services (Advarra IRB# Pro00055320) and was approved by the ICES Privacy and Compliance Officer.

## 3. Results

### 3.1. Study Population

A total of 4266 FL cases and 1763 MZL cases were initially identified. There were 126 (2.9%) FL patients and 63 (3.6%) MZL patients excluded due to invalid information. Another 3070 (72.0%) FL patients and 1402 (79.5%) MZL patients were excluded because they did not receive systemic treatment within the study period, received systemic treatment through a clinical trial, or received first-line therapy but did not receive second-line therapy. Additionally, 527 (12.4%) FL patients and 189 (10.7%) MZL patients were excluded because they received a regimen not part of the list of eligible regimens as first- or second-line therapy (Table S1, Exhibit 8).

Finally, 258 (6.0%) FL patients and 41 (2.3%) MZL patients were excluded because of refractory disease, treatment intolerance, or because they experienced transformation to DLBCL before relapse. After applying the exclusion criteria, the FL cohort included 285 (6.7%) patients who relapsed after first-line treatment and received second-line treatment. The MZL cohort included 68 (3.9%) patients who relapsed after first-line treatment and received second-line treatment.

### 3.2. Patient Characteristics

Patients from across Ontario and with first diagnosis across the study period years were included. The characteristics of relapsed patients in the FL and MZL cohorts are presented in Table 1. The median follow-up for both cohorts was 1095 days (3 years). Included patients were evenly distributed across income quintiles.

**Table 1.** Follicular lymphoma and marginal zone lymphoma relapse cohort characteristics, Ontario Canada, 2005–2018.

| Variable/Strata Name | FL Relapse Cohort | MZL Relapse Cohort |
|:---:|:---:|:---:|
| | N = 285 | N = 68 |
| **Female**, N (%) | 136 (47.7%) | 28 (41.2%) |
| **Age**, years | | |
| Mean ± SD | 62.3 ± 11.7 | 70.51 ± 10.1 |
| Median (IQR) | 63 (54–71) | 70.5 (64–77) |
| ≥65 years old, N (%) | 128 (44.9%) | 50 (73.5%) |
| **Rural residence**, N (%) | | |
| Large Urban | 193 (67.7%) | 52 (76.5%) |
| Medium Urban | 30 (10.5%) | 9 (13.2%) |
| Rural | 62 (21.8%) | 7 (10.3%) |
| **Income Quintile (Q)**, N (%) | | |
| Q1, lowest | 54 (19.0%) | 15 (22.1%) |
| Q2 | 53 (18.6%) | 8 (11.8%) |
| Q3 | 64 (22.5%) | 17 (25.0%) |
| Q4 | 54 (19.0%) | 13 (19.1%) |
| Q5, highest | 60 (21.1%) | 15 (22.1%) |
| **Local Health Integration Network (LHIN)**, N (%) | | |
| Erie St. Clair | 17 (6.0%) | * 1–5 |
| South-West | 29 (10.2%) | 9 (13.2%) |
| Waterloo Wellington | 17 (6.0%) | * 1–5 |
| Hamilton Niagara Haldimand Brant | 55 (19.3%) | 15 (22.1%) |
| Central West | * 1–5 | * 1–5 |
| Mississauga Halton | 18 (6.3%) | * 1–5 |
| Toronto Central | 11 (3.9%) | * 1–5 |
| Central | 11 (3.9%) | 6 (8.8%) |
| Central East | 30 (10.5%) | 8 (11.8%) |
| South-East | 28 (9.8%) | * 1–5 |
| Champlain | 25 (8.8%) | 10 (14.7%) |
| North Simcoe Muskoka | 11 (3.9%) | * 1–5 |
| North-East | 21 (7.4%) | * 1–5 |
| North-West | * 10–14 | 0 (0.0%) |
| **Prior cancer history**, N (%) | 6.3%) | * 1–5 |
| **First-line regimen**, N (%) | | |
| R-CHOP | 66 (23.2%) | * 4–8 |
| R-CVP | 183 (64.2%) | 54 (79.4%) |
| BR | 12 (4.2%) | * 1–5 |
| R-mono/CHOP/CVP | 24 (8.4%) | 7 (10.3%) |

**Table 1.** *Cont.*

| Variable/Strata Name | FL Relapse Cohort | MZL Relapse Cohort |
|---|---|---|
| **Second-line regimen**, *N* (%) | | |
| R-CHOP | 19 (6.7%) | * 4–8 |
| R-CVP | 8 (2.8%) | * 1–5 |
| BR | 92 (32.3%) | 27 (39.7%) |
| R-mono/CHOP/CVP/Bendamustine [#] | 115 (40.4%) | 23 (33.8%) |
| Other | 51 (17.9%) | 12 (17.7%) |
| **Time from index date to end of follow-up**, days | | |
| Mean ± SD | 804.8 ± 383.1 | 812.3 ± 386.5 |
| Median (IQR) | 1095 (459–1095) | 1095 (450–1095) |
| **Year of diagnosis**, *N* (%) | | |
| 2005 | 59 (20.7%) | 14 (20.6%) |
| 2006 | 39 (13.4%) | 6 (8.8%) |
| 2007 | 30 (10.5%) | 6 (8.8%) |
| 2008 | 26 (9.1%) | 7 (10.3%) |
| 2009 | 22 (7.7%) | 10 (14.7%) |
| 2010 | 33 (11.6%) | 9 (13.2%) |
| 2011 | 36 (12.6%) | 6 (8.8%) |
| 2012 | 40 (14.0%) | 10 (14.7%) |

**Abbreviations:** BR: bendamustine plus rituximab; CHOP: cyclophosphamide, doxorubicin, vincristine, and oral prednisone; CVP: cyclophosphamide, vincristine, and oral prednisone; FL: follicular lymphoma; IQR: interquartile range; MZL: marginal zone lymphoma; R-mono: rituximab monotherapy SD: standard deviation. [#] Combining patients receiving unique regimens to avoid small cell suppression. * Cell sizes of 1–5 were masked as per ICES SOPs.

The FL cohort was 47.7% (*N* = 136) female, with an average age of 62.3 years at the time of relapse, 6.3% (*N* = 18) had a prior history of cancer, and 67.7% (*N* = 193) lived in a large urban center. R-CVP (64%, *N* = 183) was the most common first line regimen. Only 4% (*N* = 12) of the FL relapse cohort received BR in the first line. In the second line of treatment, 40% of patients received one of R-mono/CHOP/CVP/bendamustine (*N* = 115; the exact % for each regimen could not be presented without compromising patient privacy) and a third of the FL cohort received BR (32%, *N* = 92).

The average age of patients in the MZL cohort was 70.5 years; 41.2% of the cohort was female, and 76.5% lived in a large urban center. Prior history of cancer was rare among MZL patients. R-CVP (79.4%, *N* = 54) was again the most common regimen received in the first line. BR was the most common regimen in the second line (39.7% (*N* = 27)).

### 3.2.1. Healthcare Resource Utilization

In the first year after relapse, >90% of patients in both the FL and MZL relapse cohorts made at least one visit to a GP, >80% visited an oncologist or hematologist, >90% saw other specialists, ~35% were hospitalized and >50% visited the ED (Table S1, Exhibit 6). Except for GP visits, HCRU (specialist visits, hospitalization, and ED use) decreased slightly in year 2 and year 3 following relapse in both the FL and MZL cohorts (Figure 1). On average, patients from both cohorts had approximately 10 visits to a GP in the year after relapse, which then decreased in the second year following relapse (average number of visits, FL: 7.3; MZL: 7.9), but increased again in the third year after relapse (FL: 8.6; MZL: 10). FL patients who received R-CHOP/R-CVP/BR as both first- and second-line treatment had more oncologist and hematologist visits than patients who received R-CHOP/R-CVP/BR

in the first line only. In years 2 and 3 after relapse, healthcare utilization was higher for patients who initiated third-line therapy than for those who did not across different services (Table S1, Exhibit 4 and 5).

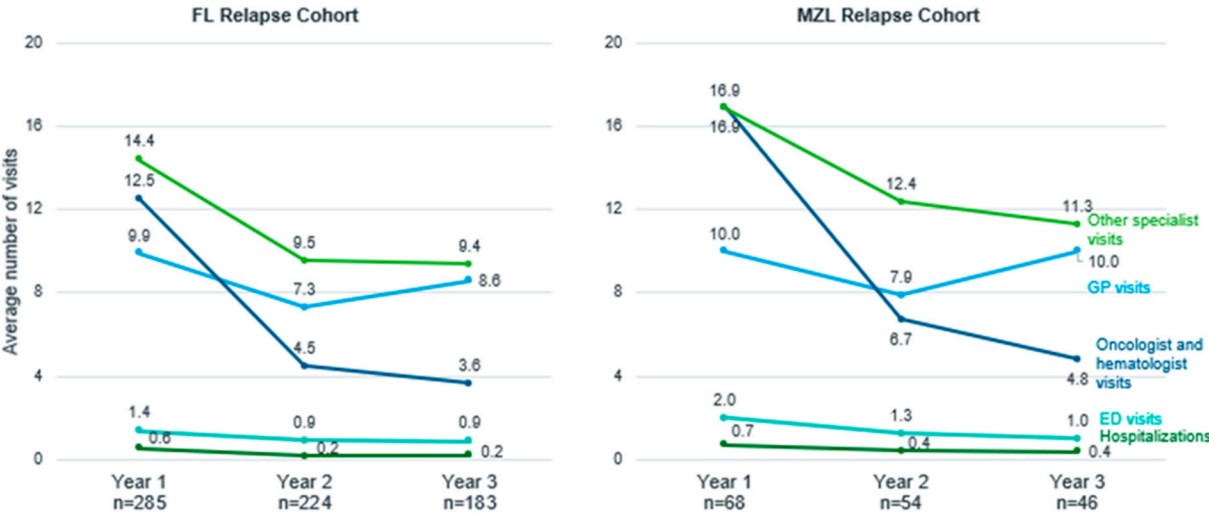

**Figure 1.** Healthcare resource utilization (average number of visits) in the three years post relapse for follicular lymphoma (FL) and marginal zone lymphoma (MZL) relapse cohorts, Ontario, Canada, 2005–2018. Abbreviations: FL: follicular lymphoma; MZL: marginal zone lymphoma; GP: general practitioner; ED: emergency department.

### 3.2.2. Direct Healthcare Costs

Nearly all patients had non-zero costs in all three years after relapse (Table S1, Exhibit 7). The average total cost of care per patient in the first year after relapse was estimated at $52,473.61 in FL patients and $59,333.81 in MZL patients. Average total costs decreased to $17,039.29 and $13,637.22 for FL patients in year 2 and year 3, respectively and to $23,396.17 and $17,300.17 for MZL patients in year 2 and year 3, respectively. Among FL patients, NDFP chemotherapy drug costs made up 35.9% ($18,863.90) of total per-patient cost in year 1 compared to 9.1% ($1241.46) in year 3, while cancer clinic costs contributed another 28.1% ($14,771.12) in year 1, compared to 17.9% ($2434.47) in year 3 (Figure 2). Similarly, among MZL patients, 34.1% ($20,209.69) of total per-patient costs were NDFP chemotherapy drug costs in year 1, compared to 7.9% ($1368.30) in year 3, and 24.1% ($14,292.38) were cancer clinic costs, compared to 12.4% ($2139.17) in year 3 after relapse.

Due to small numbers in the MZL cohort, direct healthcare costs were stratified by treatment received in the first- and second-line for FL patients only. In year 1, the average total cost for FL patients receiving rituximab-based therapy (R-CHOP/R-CVP/BR) in the first and second line was approximately $29,600 higher than that for patients receiving rituximab-based therapy in the first line only ($69,368.16 vs. $39,784.15). However, costs were lower in year 2 and 3 for patients receiving rituximab-based therapy in both lines. Among patients who received rituximab-based therapy in both the first and second line, NDFP chemotherapy drug and cancer clinic costs contributed >50% of total direct healthcare costs (Figure 3). NDFP chemotherapy drug payments contributed 44.9% and cancer clinic expenditure contributed 30.0% of total costs in the first year after relapse for patients who received rituximab-based therapy in both the first and second line compared to 23.7% and 25.6% among patients who received rituximab-based therapy in the first line only. The total average costs per patient over the three-year follow up were higher for patients with FL relapse who initiated third-line therapy than for those patients who did not with the difference increasing in year 2 and 3 (Table S1, Exhibit 7).

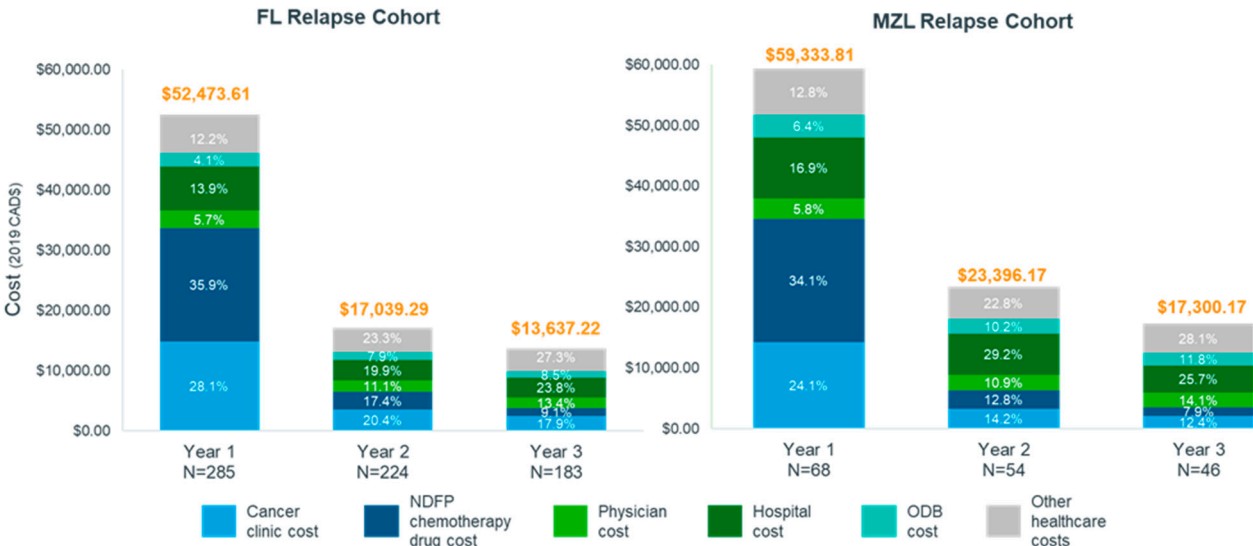

**Figure 2.** Proportion of total cost by healthcare service in the three years post relapse for follicular lymphoma and marginal zone lymphoma cohorts, Ontario, Canada, 2005–2018. Abbreviations FL: follicular lymphoma; MZL: marginal zone lymphoma; NDFP: New Drug Funding Program; ODB: Ontario Drug Benefit.

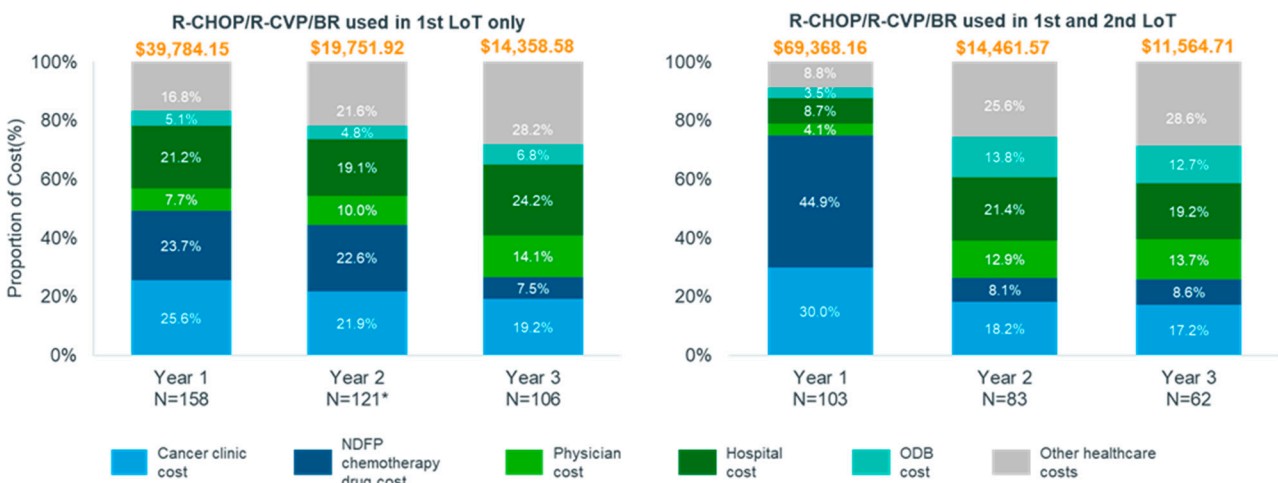

**Figure 3.** Proportion of total cost by healthcare service in the three years post relapse for the follicular lymphoma cohort stratified by treatment received in the first and second line, Ontario, Canada, 2005–2018. Abbreviations: BR: bendamustine plus rituximab; CHOP: cyclophosphamide, doxorubicin, vincristine, and oral prednisone; CVP: cyclophosphamide, vincristine, and oral prednisone; ED: emergency department; FL: follicular lymphoma; LoT: line of treatment; MZL: marginal zone lymphoma; R-mono: rituximab monotherapy; NDFP: New Drug Funding Program; ODB: Ontario Drug Benefit.

### 3.2.3. Survival Outcomes, Patient Journey, and Time to Next Treatment

Survival outcomes are presented in Figure 4. The 3-year overall survival (OS) following first relapse was 83.9% for patients in the FL cohort and 74.2% for patients in the MZL cohort. There was no statistically significant difference in survival between patients who received rituximab-based therapy in both first and second line of therapy for relapsed FL versus those who received it in first line only ($p$ = 0.30). After three years, 17.7% (95% confidence interval [CI]: 12.2%–24.1%) of FL patients on rituximab-based regimens in the first line only died compared to 13.5% (95%CI: 7.5%–21.3%) of patients who received any of R-CHOP/R-CVP/BR as both the first- and second-line treatment.

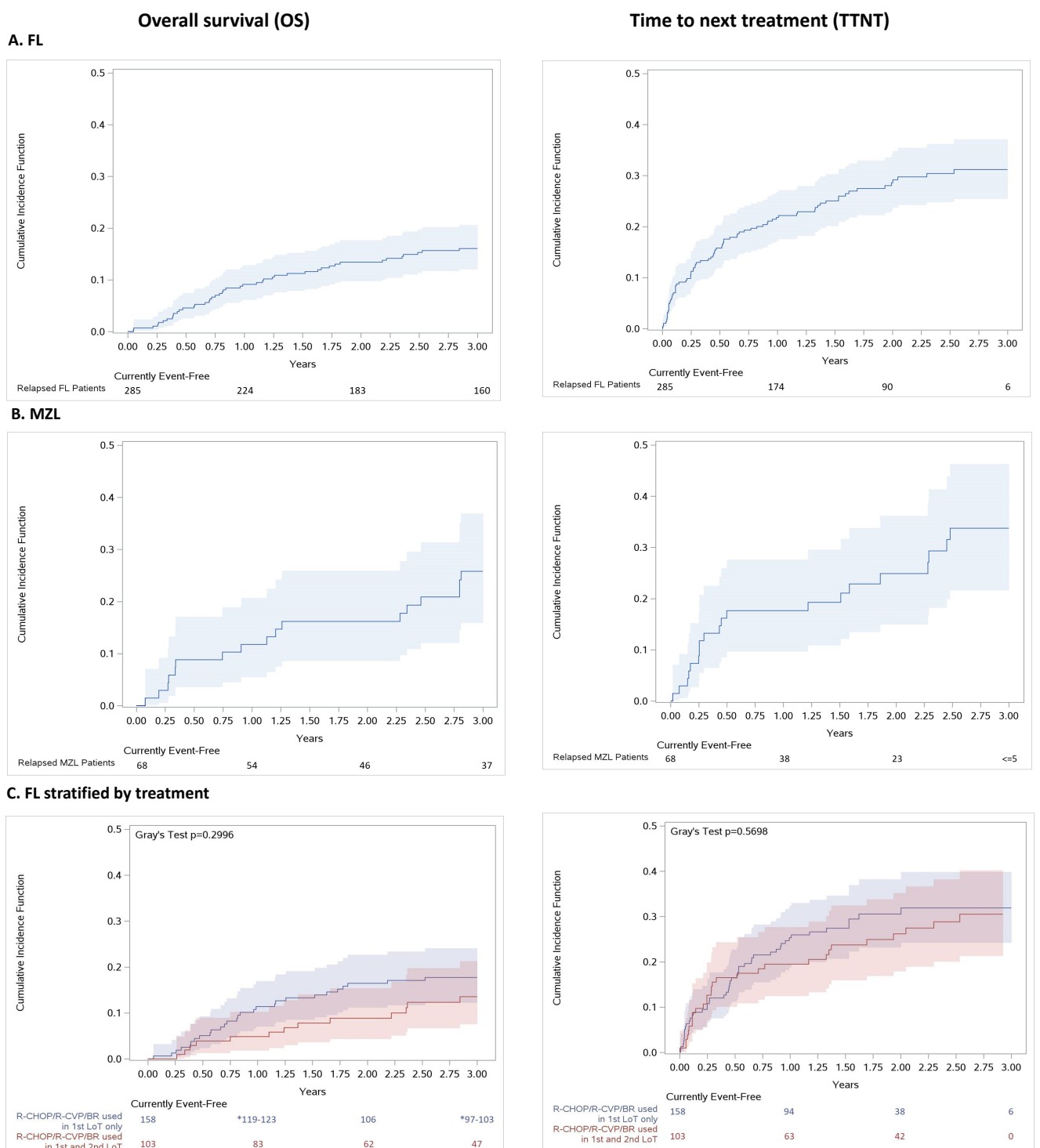

**Figure 4.** (online only). Cumulative incidence function (95%CI) of all-cause mortality and time to next treatment for (**A**) all FL relapse patients, (**B**) all MZL relapse patients and (**C**) all FL relapse patients stratified by regimen received in the first- and second-line, Ontario, Canada, 2005–2018. Abbreviations: BR: bendamustine plus rituximab; CHOP: cyclophosphamide, doxorubicin, vincristine, and oral prednisone; CVP: cyclophosphamide, vincristine, and oral prednisone; ED: emergency department; FL: follicular lymphoma; LoT: line of treatment; MZL: marginal zone lymphoma; R-mono: rituximab monotherapy.

On average, the time from diagnosis to first relapse (defined as initiating a second line of therapy for FL or MZL) was 4.3 years in the FL cohort and 4.9 years in the MZL

cohort (Table S1, Exhibit 3). FL and MZL patients initiated first-line treatment, 11.8 months and 16.1 months after diagnosis, respectively. Mean duration of first-line treatment was 12.4 months for the FL cohort and 13.4 months for the MZL cohort. Progression-free time was 24.7 in the FL cohort and 29.3 months in the MZL relapse cohort.

Approximately, 31.2% (95% CI: 25.4%–37.1%) of patients in the FL and 33.7% (95% CI: 21.6%–46.3%) in the MZL relapse cohorts initiated a third line of therapy during the study period. For those individuals who initiated the next therapy, the average time from the second-line of therapy to the third-line of therapy was 7.6 months for the FL cohort and 11.2 months for the MZL cohort (Figure 4A,B; Table S1, Exhibit 3). No statistically significant difference was observed in TTNT between patients who received rituximab-based regimens in the first line only versus those patients who received rituximab-based therapy in both the first and second line (*p* = 0.57; Figure 4C).

## 4. Discussion

This study used administrative data from the province of Ontario, Canada, to estimate HCRU, direct healthcare costs, and outcomes in patients who relapsed following initial treatment for FL and MZL. In general, trends in healthcare use and patient outcomes were similar between FL and MZL. Most patients received treatment in line with recommendations made by the Lymphoma Canada Scientific Advisory Board guidelines [10]. For patients who relapsed after initial therapy, the duration of response for second-line treatment was shorter, suggesting that the interval of time to relapse decreases with each subsequent relapse event, which is a pattern that was previously noted in the literature [20,21]. Survival after relapse in both cohorts was high, consistent with what is reported in the literature [22] and did not vary significantly for FL patients who received a rituximab-based regimen in the second line of treatment and those who did not.

For both the FL and MZL cohorts, resource use and costs were highest in the year immediately following relapse and decreased in subsequent years due to decreasing drug and cancer clinic costs. This was consistent for FL patients who did and did not receive rituximab-based therapy in the second line of treatment. Given that most patients were treated with rituximab-based regimens, the cost of rituximab and its administration may be contributing to the higher NDFP drug and clinic costs in the year after initial relapse. Fowler et al. 2020 also reported that drug spending made up the highest proportion of annual costs across treatment lines [15]. Chemotherapy and pharmacy costs of USD100,000 to USD425,000 constituted the major proportion of the annualized cost of treatment lines. Higher HCRU in the years following relapse can be explained by chemotherapy-related toxicity and treatment resistance, which represent challenges in treating patients in the second line and beyond. The overall, mean per-patient, per-month cost during 2 years of follow-up was USD10,460. Furthermore, patients requiring subsequent lines of treatment were likely to be older and have more comorbid conditions than newly diagnosed patients. Disease progression was also a major driver of HCRU, cost and health-related quality of life following relapse after initial treatment and associated with substantial economic burden [23]. Beveridge et al., in a USA study, reported a four-fold increase in annual costs and medical visits of $30,890 for progression versus $8704 for non-progression [24].

While costs decreased in years 2 and 3 after relapse, they would be expected to increase again during subsequent relapse events, which are likely to occur at shorter intervals as patients become more refractory to treatment. Additionally, with increasing lines of treatment, the likelihood of accumulating other medical problems and treatment-related toxicities that lead to hospitalization also rises, leading to a greater need for management of symptoms and psychosocial support, which also results in increased visits to a general practitioner/family physician.

The use of administrative data with complete billing information for the main publicly reimbursed health services received is a major strength of this study. Administrative data allows for large sample sizes and produces evidence that is generalizable and applicable for policy. Relapsed FL and MZL are typically understudied due to their indolent nature

and the long follow-up required to observe key health outcomes. The availability of retrospective administrative data allowed for the assessment of the burden of illness based on a relatively high number of patients with comprehensive data over a longer period.

This study also had some limitations. First, the final sample for this analysis constitutes 6.7% and 3.9% of all FL and MZL patients, respectively, who were newly diagnosed between 1 January 2005 and 31 December 2012. Most patients were excluded because they did not receive systemic treatment, or completed first-line therapy but there was no record of second-line treatment, or received treatment on a clinical trial (72.0% of FL and 79.5% of MZL patients). Another study of 727 FL patients in Barcelona, Spain, reported that 3–30% of patients treated between January 1980 and December 2017 did not receive systemic therapy ("watchful waiting" strategy), 4–10% had refractory disease, 7–18% experienced transformation to DLBCL, and 19–46% progressed within two years of initial treatment [25]. In general, indolent lymphomas comprise 35–45% of NHL, while approximately 20% of FL patients require treatment [26,27]. Second, despite excluding patients with refractory disease, treatment intolerance, or patients who experienced transformation to DLBCL, this analysis was enriched for patients who had more aggressive FL or MZL disease, were likely to experience early second-line failure and excludes patients who had longer remission times. These criteria were applied because we looked to study the costs associated with second-line treatment and, thus, excluded patients that continued in remission following primary treatment. While this excludes patients that may go on to experience later relapses, there will be heterogeneity of outcome based on disease and treatment characteristics [28]. We also did not examine those who had stem cell transplants. While we observed a small number of individuals in our study receiving stem cell transplantation, the small sample size prevented their inclusion.

It is also important to note that the landscape for the treatment of FL and MZL in Canada changed over the study period and changed further since. Intravenous rituximab in combination with CVP was only approved for use as first-line therapy in the treatment of NHL by Health Canada on 20 December 2005 and as maintenance treatment on 28 July 2006, part way through the study period. Bendamustine in combination with rituximab received a Health Canada notice of compliance for use in the relapsed setting in August 2012 and public reimbursement in Ontario in May 2013 and then moved into use in the first-line setting [29]. In 2018, a subcutaneous preparation of rituximab was approved as a fixed dose for combination therapy, simplifying administration and reducing systemic therapy suite time [8], with a corresponding reduction in costs related to administration. Additionally, rituximab biosimilars for the treatment of NHL were approved in Canada in 2021, which may reduce the cost of treatment going forward [30]. Nevertheless, this study provided a comprehensive baseline estimate of the economic burden of relapsing FL and MZL to which the cost of emerging treatments and combinations can be compared.

More recently, the next generation anti-CD20 monoclonal antibody obinutuzumab (O) was studied in combination with CHOP and bendamustine in patients with FL who relapse within 2 years after first-line treatment with chemotherapy and anti-CD20 therapy, including O-umralisib (Ukoniq), and O-lenalidomide [31–33]. Emerging treatment options for relapsed/refractory disease also include ibritumomab tiuxetan, lenalidomide with rituximab, umbralisib, ibrutinib, and tazemetostat (Tazverik) [13]. Newer therapies including cellular therapy, bispecific antibodies, and ibrutinib (relapsed/refractory MZL only) improved outcomes in patients who were rituximab refractory at the time of relapse [12,13]. Combination therapies that result in better disease control (than rituximab monotherapy) with more manageable toxicity (than rituximab and chemotherapy) [15] and that delay or prevent progression may reduce mortality and morbidity. Furthermore, while these new agents will likely increase drugs costs, it is unclear how these drugs may affect other aspects of health care resource utilization related to FL and MZL given the varied means of administration and different toxicity profiles [24].

## 5. Conclusions

This study provided a comprehensive estimate of health care resource utilization and costs associated with the treatment of relapsing FL and MZL, based on currently available treatments in Canada. Relapsed FL and MZL represent a significant burden to the patient and healthcare system, particularly considering these are regarded as indolent cancers. Interventions that delay progression may lead to savings in healthcare costs and improve patient outcomes.

**Supplementary Materials:** The following supporting information can be downloaded at: https://www.mdpi.com/article/10.3390/curroncol30050352/s1, Table S1: ICD-O-3 morphology codes for follicular lymphoma and marginal zone lymphoma.

**Author Contributions:** Conceptualization, J.K., E.M.E., J.E.-P., R.N., A.K. and A.S.; formal analysis, R.N. and M.E.; funding acquisition, E.M.E. and J.E.-P.; investigation, R.N. and M.E.; methodology, J.K., E.M.E., J.E.-P., R.N., M.E., A.K. and A.S.; project administration, A.K. and A.S.; resources, A.K. and A.S.; supervision, E.M.E., J.E.-P., A.K. and A.S.; validation, R.N. and M.E.; writing—review and editing, J.K., E.M.E., J.E.-P., R.N., M.E., A.K. and A.S. All authors have read and agreed to the published version of the manuscript.

**Funding:** This study was funded by Janssen Inc.

**Institutional Review Board Statement:** The study was conducted in accordance with the Declaration of Helsinki and was approved by the Institutional Review Board (or Ethics Committee) of Advarra IRB (protocol code Pro00055320 and 3 November 2020).

**Informed Consent Statement:** The use of data in this project was authorized under Section 45 of Ontario's Personal Health Information Protection Act and, as a result, informed consent was not required.

**Data Availability Statement:** The dataset from this study is held securely in coded form at ICES. While legal data sharing agreements between ICES and data providers (e.g., healthcare organizations and the government) prohibit ICES from making the dataset publicly available, access may be granted to those who meet pre-specified criteria for confidential access, available at www.ices.on.ca/DAS (accessed on 1 March 2023) (email: das@ices.on.ca). The full dataset creation plan and underlying analytic code are available from the authors upon request, understanding that the computer programs may rely upon coding templates or macros that are unique to ICES and are, therefore, either inaccessible or may require modification.

**Acknowledgments:** Thank you to Nancy He, Jacob Etches, and Jenna Novess of ICES for data management and analytic support. This study made use of de-identified data from the ICES Data Repository, which is managed by ICES with support from its funders and partners: Canada's Strategy for Patient-Oriented Research (SPOR), the Ontario SPOR Support Unit, the Canadian Institutes of Health Research, and the Government of Ontario. The opinions, results, and conclusions reported are those of the authors. No endorsement by ICES or any of its funders or partners is intended or should be inferred. We acknowledge support from Asad Husain, Brendan Osborne, and Neerav Monga of Janssen Inc. for strategic input into this study. Medical writing support was provided by Suchitra Jagannathan and Lidija Latifovic of IQVIA.

**Conflicts of Interest:** Emmanuel Ewara is an employee of Janssen Inc. and a stockholder of Johnson & Johnson. Julia Pacitti is an employee of Janssen Inc. and a stockholder of Johnson & Johnson. Ryan Ng, Maria Eberg, Atif Kukaswadia, and Arushi Sharma are employees of IQVIA Solutions Canada Inc. IQVIA is a contract research organization which received consulting fees from Janssen Inc.

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
