# Peer review of "Estimating the Burden of Illness of Relapsed Follicular Lymphoma and Marginal Zone Lymphoma in Ontario, Canada"

_curroncol, doi:10.3390/curroncol30050352_

Round 1

Reviewer 1 Report

I find the subject matter interesting and, overall, all the work. I make a few considerations that I would like the authors to evaluate. First of all, I do not find the bibliographic reference n.5 adequate. Than, there is no mention of subjects undergoing stem cell transplantation. It would be correct to cite the reasons for this. Finally, there are no data on comorbidities, although they are mentioned at the beginning. Specify better how they were used. 

Author Response

I find the subject matter interesting and, overall, all the work. I make a few considerations that I would like the authors to evaluate. First of all, I do not find the bibliographic reference n.5 adequate. Than, there is no mention of subjects undergoing stem cell transplantation. It would be correct to cite the reasons for this. Finally, there are no data on comorbidities, although they are mentioned at the beginning. Specify better how they were used.

Response:

We thank the reviewer for their comments. In response:

  • Reference 5: Two references were added to address this comment (Lines 47-49); Links provided here for brevity: https://pubmed.ncbi.nlm.nih.gov/26989206/ and https://pubmed.ncbi.nlm.nih.gov/35139275/
  • Stem cell transplant: We examined this, however, did not present these findings as very few patients had a stem cell transplant in the follow-up period. This corresponded to 29 (~10%) FL patients and less than 5 MZL patients. This would have made the interpretation of this a challenge due to the small samples sizes of patients who received stem cell transplants, combined with the ICES policy of suppressing any cells smaller than 6. This was added to the discussion (Lines 376-378).
  • Comorbidities: We used this to profile the FL and MZL patients, however, these results were not presented in the final manuscript for brevity. This clause was removed from the revised manuscript (Line 123)

Reviewer 2 Report

This is a comprehensive study on health resource utilization for the treatment of relapsing follicular lymphoma and marginal zone lymphoma - two indolent and slow-growing subtypes of non Hodking lymphoma. Additionally,  treatment patterns, progression and overall survival of patients in Ontario, Canada is provided.  The introduction gives sufficient background for this study, underlining that the literature on this important financial aspect of patients care  is scarce. Materials and Methods section provides detailed description of study design. Noteworthy, the criteria for inclusion of patients into study are very rigorous making sizes of groups relatively small. The statistical methods employed are appropriate. Results are clearly presented and thouroughly described. The discussion is well written and inspiring. 

Minor concerns:

Figure 1 (GP and ED abbreviations should be explained in the figure caption and not only in the main text)

Figure 4 (the font size describing numbers of patients below graphs is too small to be visible without magnification. If possible, please increase the font size)

Author Response

This is a comprehensive study on health resource utilization for the treatment of relapsing follicular lymphoma and marginal zone lymphoma - two indolent and slow-growing subtypes of non-Hodgkin lymphoma. Additionally,  treatment patterns, progression and overall survival of patients in Ontario, Canada is provided.  The introduction gives sufficient background for this study, underlining that the literature on this important financial aspect of patients care  is scarce. Materials and Methods section provides detailed description of study design. Noteworthy, the criteria for inclusion of patients into study are very rigorous making sizes of groups relatively small. The statistical methods employed are appropriate. Results are clearly presented and thouroughly described. The discussion is well written and inspiring.

Response: We thank the reviewer for the comments and feedback, and appreciate the support for the manuscript.

Minor concerns:

Figure 1 (GP and ED abbreviations should be explained in the figure caption and not only in the main text)

Response: These were added to the figure caption

Figure 4 (the font size describing numbers of patients below graphs is too small to be visible without magnification. If possible, please increase the font size)

Response: We have updated the caption in Figure 1. For Figure 4, this was a consequence of this being condensed to one page; the revised Figure 4 is clearer.

Reviewer 3 Report

This is an interesting overview of costs in a population of R/R indolent lymphomas. The analysis is well done and provides interesting knowlwdges on healthcare costs, with which we will increasingly interface, in particular giving the expected widespread use of new drugs.

Just a couple of minor comments/questions:

- I wold suggest to not mention drugs commercial names

- Which treatment time-period has been considered for the analyasis? (only the years of the diagnosis, but not years of treatment are reported)

- Most of patients received in first line R-CVP that is not anymore a standard of care, expecially in follicular lymphoma. At present BR is for sure preferred first line, at least in MZL and the role of chemoimmuno in R/R is pregressively reducing

- The large availability of biosimilar rituximab should significantly lower the expected costs, authors should mention t and comment the costs that are expected from the use of biosimilars 

- do the authors have some info on specific costs related to infective complications (e.g. hospitalizations, CT scan, iv Ig prophylaxis...)?

Author Response

This is an interesting overview of costs in a population of R/R indolent lymphomas. The analysis is well done and provides interesting knowledges on healthcare costs, with which we will increasingly interface, in particular giving the expected widespread use of new drugs.

Response: Thank you for the feedback on the manuscript.

Just a couple of minor comments/questions:

- I would suggest to not mention drugs commercial names

Response: We have removed the commercial names of the drugs.

- Which treatment time-period has been considered for the analysis? (only the years of the diagnosis, but not years of treatment are reported)

Response: Patients who were newly diagnosed with FL/MZL between January 1 2005 and December 31 2012, and who relapsed before December 31 2018 were identified (Lines 99-101). They were followed for up to three years post-relapse, and were censored at the first of death, censoring due to histological transformation to diffuse large B-cell lymphoma (DLBCL), end of the 3-year study period, end of Ontario Health Insurance Plan (OHIP) coverage, or March 31, 2020, whichever came first (Lines 101-104). Therefore the patients would have different treatment time periods depending on their date of relapse.

- Most of patients received in first line R-CVP that is not anymore a standard of care, especially in follicular lymphoma. At present BR is for sure preferred first line, at least in MZL and the role of chemoimmuno in R/R is progressively reducing

Response: We thank the reviewer for this comment and agree with their insight. Over the course of our study analysis period (2005 to 2020) there have been significant advances in the treatment of FL/MZL. We mention this as a limitation of the study in the discussion (lines 379-389).

- The large availability of biosimilar rituximab should significantly lower the expected costs, authors should mention t and comment the costs that are expected from the use of biosimilars

Response: We agree with this comment and cost would have changed with the introduction of biosimilars in this market in 2021. This may result in a reduction in the costs of treatment, and we mention this in our discussion (Lines 389-390).

- do the authors have some info on specific costs related to infective complications (e.g. hospitalizations, CT scan, iv Ig prophylaxis...)?

Response: Unfortunately, these data were not captured as part of this study. This would be a very interesting follow-on study however, and we will take this back to the team for consideration for future work.